# A Critical Analysis of Agile and Lean Methodology to Fulfill the Project Management Gaps in Nonprofit Organizations (NPOs)

Fares AbuKhamis * and Abdelhakim Abdelhadi

College of Engineering, Prince Sultan University, Riyadh 11586, Saudi Arabia; abdelhadi@psu.edu.sa
* Correspondence: 219120716@psu.edu.sa

**Abstract:** This research examines the implications of lean and agile methodologies for solving project management gaps in nonprofit organizations (NPOs). The data were collected from project managers, executives, and supervisors through an online survey questionnaire. The value of Cronbach's Alpha is below 0.8, indicating reliable, valid, and consistent results. This study also concludes that the primary cause of the incapability of the previously mentioned methodologies could be the lack of awareness from project respondents because only 40% and 36% knew about lean methodology and the agile approach, respectively, which could be a significant drawback. The findings of this study state that Agile and Lean methodology implantation in project management allows the recording of progress and teamwork performance periodically, which helps monitor projects that are going as per planned objectives or where remedial actions are required to be undertaken. Moreover, the agile technique should be employed in a project where the role of managers is crucial in this regard because sensitive issues, such as cost and efforts require constant monitoring and urgent attention. Based on the present study's results, a hybrid strategy is necessary, combining Agile and Lean approaches to examine diverse business situations. Researchers and project experts need to add more factors to measure project gaps accurately and explain how Agile and Lean management work to change the success of a project.

**Keywords:** non-profit organization; lean manufacturing; agile methodology; project management

## 1. Introduction

The research objective is to evaluate the use of the lean and agile methodology and highlight the implications for addressing the project management gaps within non-profit organizations (NPOs). NPOs must make the most of their scarce resources due to everyday challenges, such as insufficient funding, an inadequate number of workers, and a lack of adequate infrastructure. This observation illustrates the importance of having a well-functioning management solution in non-governmental organizations (NGOs). Managers, members of the team, and consumers can all benefit from the agile methodology. Necessary measurements, such as turnaround time, order cycle, and productivity are going to be identified by agile methodology, which might be employed to examine a team's productivity, identify potential problems, and create data-driven strategies to solve them. Non-profit organizations range in size from small to large, local to global, and are operated by staff or volunteers. Agile and Lean are widely used methods for quality improvement and business management. An agile approach is used to implement and evaluate product development methods, while lean strategies aim to remove waste in any system. Eliminating unnecessary activities reduces effort and gives managers greater discretion. Providing a service or product from an idea to a customer or end-user needs value sharing. The adoption of Agile and Lean methodologies requires managerial adjustments. Systems, such as healthcare, education, insurance, and financial sectors adopted lean organizational methods in the early 1920s [1]. Agile and lean procedures are now part of a single performance management system. Agile and lean methodologies may be applied to improve any product or service process.

Smaller non-profit organizations, NPOs, must implement faster and smoother sustainable approaches to survive. While Governments increasingly depend on donors to provide effective help to these organizations, small charities must constantly innovate to survive. A lack of government support, declining private funding, and increased rivalry from Non-Governmental-Organizations, NGOs and other profit-making enterprises might endanger an NPO's survival. Some non-profit executives limit operating funding. Non-profits may need varying degrees of growth. Examine the organization's purpose and aims. This requires an in-depth grasp of end-user preferences and demands, such as what the customer wants.

This research discusses the usage of Agile and Lean methods in non-profit organizations. The agile method refers to regularly reviewing the project's primary requirements or product. Agile in a non-profit requires the lowest possible product. Agile encourages direct project management and simplicity. This study will assess, explain, and analyze the anticipated effort of adapting lean and agile project management systems to NPOs. A guide on using Agile and Lean approaches in non-profit organizations will also be provided as part of this research.

## 2. Literature Review

Glover and Hurley [2] argue that reduction in cost and productivity improvements have become one of the primary priorities for each organization in the contemporary era of highly competitive marketplaces. These difficulties must be solved not only by commercial enterprises but most importantly, by non-profit organizations (NPOs). Furthermore, Sońta-Drączkowska and Mrożewski [3], in their analysis, have shown that firms worldwide are adopting agile management approaches more than the waterfall methodology management practices to increase long-term advantages, particularly in non-profit organizations. Sodhi and Singh's [4] analysis has revealed the importance of agile management methods and their positive effect on the effectiveness of a non-profit corporation.

Nahrkhalaji et al. [5] and Anderson and Lannon [6], stated that NPOs face issues and challenges due to limited resources and capital. They often turn their attention to improving their strategies and procedures for efficiency and effectiveness. It is indicated that the primary objective of Non-Profit Organizations is to allocate scarce resources in an efficient manner that brings out maximum utilization and minimum cost for the services of the deserving community [2]. Non-Profit Organizations have a broad supply chain network and logistic chain to serve efficiently and effectively [7]. It is supported by Kashikar et al.'s [8] research, that 80% of the Non-Profit budget is utilized in the Logistics and Supply Chain network, which is a questionable percentile. The supply chain and logistic network is the most crucial operation for a non-profit organization, which requires implementing an efficient and innovative model to effectively procure goods, ensure timely delivery of services, and safeguard resources [9]. However, the non-profit organization failed to do so, due to inadequate knowledge, lack of qualified human resources, insufficient resources, and slow adoption of technology. The agile method refers to the continuous reexaminations of the project's basic requirements or the product and making the changes required by the project [10]. Concerning the application of the agile method in a non-profitable organization, it is necessary to describe the least possible viable product. After that, bringing together a cross-functional group and motivating the group members [11]. According to the research study conducted by Obradović, Todorović and Bushuyev [12], agility denotes the state of mind while an individual deals with a project. Agile methodology in project management prioritizes easiness, flexibility, and face-to-face interaction. The lean method and its practices assist NPOs in rectifying and spotting the bottle, which saves the organization money, energy, and time. However, it also depends on the bottleneck, as many things can be performed to address it [13]. As per the lean methodology, all those things that do not contribute to manufacturing the products are waste materials. The lean waste is transport, inventory, skills, waiting, useless motion, overproduction, over-processing, and defects [14]. Waste due to transportation means people, inventory, tools, and equipment

without any purpose. Extreme movement of equipment and tools causes damage and defects in them. Furthermore, people's movements can cause exhaustion and pointless work [15].

Another reason is that the NPOs seem to lack trust in innovative and contemporary models due to the lack of financial support and stakeholder interference as it becomes difficult for NPOs to obtain money, donations, and charities to experiment with new models and practices [16].

According to a recent Forbes Insights and Scrum Alliance survey [17], end-to-end Agile and Lean businesses reduce market times, increase revenue, and foster innovation. Nonprofits may need varying degrees of growth. Examine the organization's purpose and aims. This requires an in-depth grasp of end-user preferences and demands. Input from customers is required [18]. The agile method refers to regularly reviewing the project's primary requirements or product [18]. Agile in a non-profit requires the lowest possible product. Then inspire the cross-functional group [19]. Although Agile and Lean ideas date back to the late 1860s, methodologies, attitudes, and development frameworks emerged in the 1980s [20].

Agile encourages direct project management and simplicity, while lean focuses on eliminating waste [21]. Less waste means more money, less energy consumption, and time availability for NPOs [22]. Individuals may be assisted by the employees of non-profit organizations. This requires learning to adapt fast to changing conditions and reaching more people with fewer resources [23]. Agile and lean approaches are favored by employees because they simplify work and accelerate processes [24].

## 3. Research Design

This research has followed a quantitative research design. This is due to the scope of this research, which is explanatory and empirical, as the researcher aims to examine how the use of the lean and agile methodology in projects could allow project managers to overcome project management gaps in NPOs. Furthermore, the quantitative design of the present research study is deliberated to be more applicable and suitable in this context as it is more objective, scientific, and focused [25].

Moreover, quantitative research design analyzes the scale of the impact of the independent variables on the dependent variables. A straightforward way of the research study outcomes is given in this form of research [26]. The research requires collecting data from the Likert scale for identified constructs in other analyses. Then statistical tests are applied to analyze the cause and effect [27]. This makes this research empirical where quantitative design seemed suitable. In terms of the research approach, this further sets the research study path being conducted. Researchers adopt two types of research approaches while conducting the research study. Moreover, those are the deductive research approach and inductive research approach. By comparing both the systems, the critical difference between them is the time frame in which the hypothesis is being made. In the current study, the deductive approach has been applied because the investigator, by using this approach, can develop conceptual models and tests by using quantitative research design and move from general focus and theories to specific focus research questions and objectives. The reason for selecting this research approach is that the current research is being conducted to add to the available literature. The current policy has some benefits: the first benefit is that it has the prospect of clarifying the relationship between different concepts. Furthermore, it can determine the perceptions from the quantitative approach and the potential of simplifying the outcomes of the research study to a certain point. The following paragraph presents the elements used to conduct the surveys.

### 3.1. Participants

The data were collected using a questionnaire survey distributed to managers and industry experts to collect the data in this industry. This data collection was conducted online, while 97 individuals took part in this study.

### 3.2. Measures

The main goal of this survey was to learn Agile and Lean methodology, to fill project management vacancies in NPOs. The study used quantitative data that were analyzed statistically. The variability of the questionnaire was measured using checklists and the Likert scale. The questionnaire was divided into two sections, each with different questions to examine critical factors within the lean and agile methodology that influences project management. The entire survey was conducted by adopting past questions from past research.

### 3.3. Method

A comprehensive survey was conducted to assess the efficacy of lean and agile approaches in improving NPO effectiveness and implementing management commitment in the organizational component of the agile process. The research shows that randomizing questions helps reduce bias in assessing ideas and themes. As a result of this, the survey randomized questions on Agile and Lean techniques. The poll asked about NPO project management and Agile and Lean approaches. Aspects of the lean and agile approach that impact project management and consequences in resolving project management deficiencies within NPOs.

### 3.4. Statistical Analysis

The data were gathered from secondary sources while the researchers gathered the original data. This study used a quantitative, explanatory, and empirical research approach to evaluate how using lean and agile techniques in projects might help NPOs overcome project management gaps. This incorporated material previously collected and analyzed by other researchers [12]. Secondary data may be less precise, yet provide valuable information. Obtaining original data may be challenging at times; secondary research is easier and more realistic in these cases. The *t*-test was also used with the Minitab Statistical Package to look at how well lean and agile project management techniques work for men and women.

## 4. Results

This section has been developed to analyze the study results more comprehensively. The methodology that has been used in the study is primarily quantitative. However, different tests are used to analyze the results, including frequency, correlation, and independent *t*-tests. Moreover, this section has assisted in the development of the next section and has helped provide recommendations effectively. The results of this study indicate that there is a significant relationship between Agile and Lean methodologies. Resolving project management gaps in the NPOs, the results of statistical model correlation analysis evidence that agile methodology helps determine progress tracking, commitment, and related implementation gaps in project management.

On the other hand, the lean methodology can resolve equipment and resource management gaps. The present study also provides the pathway to comprehending agile approaches in the project management of NPOs. The independent *t*-test and the One-way ANOVA results further indicate no significance in the model. From a conceptual perspective, the present study focused on one variable: project management gaps in NPO to elucidate the implications of lean and agile approaches to deal with these issues and enhance project performance.

Several statistical techniques have been applied to generate the results and were used in the sequence of demographics analysis, where the type of respondents in this study is analyzed. First, a test of reliability analysis is conducted to examine the internal consistency, reliability, and validity of the responses generated. It is followed by correlation analysis to identify the relationship between project management methodology and resolving project management gaps. A *t*-test is autolyzed to check if there is any difference of opinion related to the effectiveness of Agile and Lean methods in determining identified project management gaps in NPOs.

### 4.1. Demographics or Type of Respondents

Table 1 indicates that 64.42% of the respondents are aged 20 to 30, whereas 25% are 31 to 40. The table also shows that 8.65% of the respondents are in the age group of 41 to 50, whereas 0.96% of the respondents are in the age group of 51 to 60, and the remaining respondents are in the age group of 61 to 70. Table 1 indicates that the idea behind collecting data from moderate to higher age respondents is that they represent more experience in the field. This criterion has been attained as most of the study participants are between 30 to 40 years of age.

**Table 1.** Age of respondents.

| Age | Count | Percent | Compact |
|---|---|---|---|
| 20–30 | 67 | 64.42 | 64.42 |
| 31–40 | 26 | 25 | 89.42 |
| 41–50 | 9 | 8.65 | 98.08 |
| 51–60 | 1 | 0.96 | 99.04 |
| 61–70 | 1 | 0.96 | 100 |

Table 2 indicates that 25% of respondents are females, and 79% are males. This shows that the majority of the respondents are males.

**Table 2.** Gender of respondents.

| Gender | Count | Percent | Compact |
|---|---|---|---|
| Female | 25 | 24.04 | 24.04 |
| Male | 79 | 75.96 | 100 |

Table 3 indicates that 55.7% of the respondents have experience of fewer than five years, whereas 19.23% of the respondents have experience of 5 to 10 years. Table 3 also shows that 96% of the respondents have experience of 21 to 30 years. Experienced participants were recruited to obtain more informed and valid answers. Therefore, Table 3 signifies that most respondents are experienced from 10 to 30 years of presenting knowledge and expertise in the field. The accumulated percentage of experienced respondents is 40.39%.

**Table 3.** Working experience of respondents in their relevant sectors.

| Years of Experience | Count | Percent | Compact |
|---|---|---|---|
| <5 | 58 | 55.77 | 55.77 |
| 5 to 10 | 20 | 19.23 | 75 |
| 10 to 15 | 10 | 9.62 | 84.62 |
| 16 to 20 | 12 | 11.54 | 96.15 |
| 21 to 30 | 3 | 2.88 | 99.04 |
| 31 and above | 1 | 0.96 | 100 |

Table 4 indicates that 65.38% of the respondents are unaware of agile methodology, whereas 34.62% have stated methods.

**Table 4.** Do you know about agile methodology?

| Do You Know about Agile? | Count | Percent | Compact |
|---|---|---|---|
| about Agile Method | | | |
| No | 68 | 65.38 | 65.38 |
| Yes | 36 | 34.62 | 100 |

Table 5 indicates that 61.54% of the respondents are unaware of the lean methodology, whereas 38.46% are aware of the lean methods.

**Table 5.** Do you know about lean methodology?

| Do You Know about Lean? | Count | Percent | Compact |
|---|---|---|---|
| No | 64 | 61.54 | 61.54 |
| Yes | 40 | 38.46 | 100 |

### 4.2. Reliability of Experiment

In this test, internal consistency and reliability have been analyzed to see if the data collected are fit and consistent for further analysis, ensuring the reliability and validity of the results generated in this study. This test is carried out by applying Cronbach's Alpha test, where the value should be higher than the threshold value, which is 0.77 as seen in Table 6. Items under every construct receiving 0.77 or higher alpha values are reliable. The Cronbach's Alpha of agile is 0.77, indicating that the study's variables are reliable and consistent. This reveals that the data under this construct is valid, consistent, and dependable for further analysis.

**Table 6.** Agile Cronbach's value.

| Cronbach's Alpha |
|---|
| 0.7749 |

The Cronbach's Alpha, shown in Table 7, for lean methodology is 0.69; this indicates that the study variables are reliable and consistent as the threshold for Cronbach's alpha should be above 0.69, which means that the value received is almost 0.7. This signifies that the data under this construct are also reliable, valid, and consistent. Therefore, it can be relied upon for further analysis.

**Table 7.** Lean Cronbach's value.

| Alpha |
|---|
| 0.6992 |

### 4.3. Frequency Analysis

Figure 1 demonstrates that 80.55% of respondents believe that management's commitment to the agile process's organizational component has an influence on the overall estimated cost and effort. At the same time, 19.44% are undecided, indicating that the agile approach's commitment has an impact on costs and efforts. This demonstrates that under agile methodology, increasing the amount of accountability for efficient project management reduces costs and improves overall project results. Figure 2 shows that 80.55% of respondents agreed with the assertion that process tracking in the agile approach had an influence on product delivery, while 13.89% were undecided. The remaining 5.56% are opposed to the idea. So, management might be able to keep a better eye on how the project is going if they use an agile method that makes it easy to spot mistakes and fix them quickly.

Figure 3 shows that 72% of respondents believe that the agile approach guarantees that project planning and design generate successful outputs. The remaining 5.56% disagreed with the notion that the agile approach delivers practical outcomes, while 11.11% remained indifferent. As a result, the gap between designing and planning a project well before putting it into action may be filled by an agile approach, which reduces the likelihood of disputes during project management.

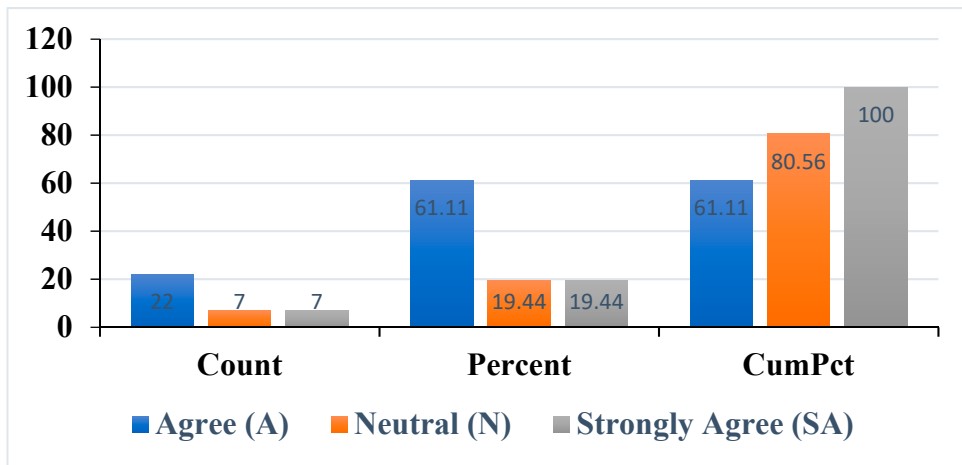

**Figure 1.** Frequency analysis.

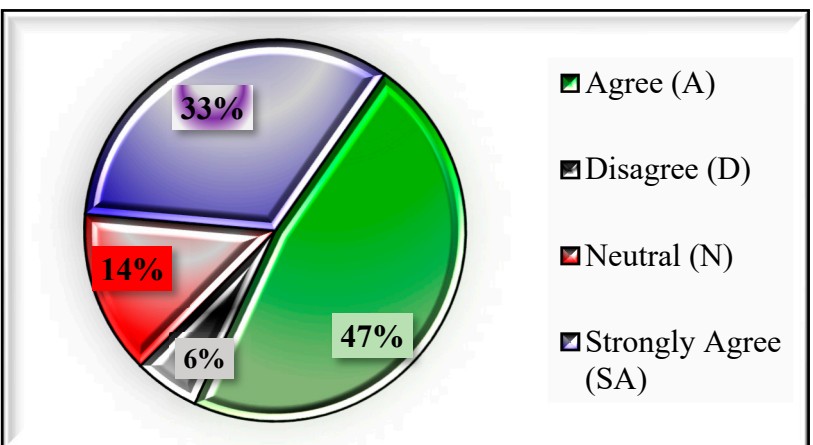

**Figure 2.** Progress tracking with agile methodology.

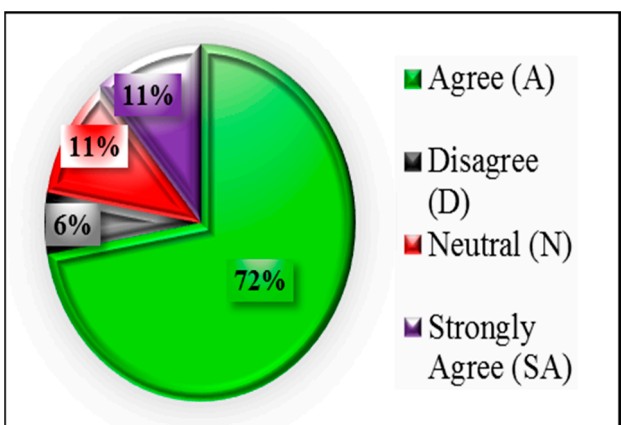

**Figure 3.** Proper planning and designing with agile methodology.

According to Figure 4, 63.89% of respondents agreed with the assertion that the participation of the customer in the agile process influences the overall anticipated cost and effort, while 27.78% were undecided. The remaining 8.33% disagreed with the assertion, implying that the client's presence influences costs and efforts. This indicates that the agile methodology provides customer presence throughout the project's design, planning, and execution, resulting in client engagement in project deliverables and ensuring client satisfaction.

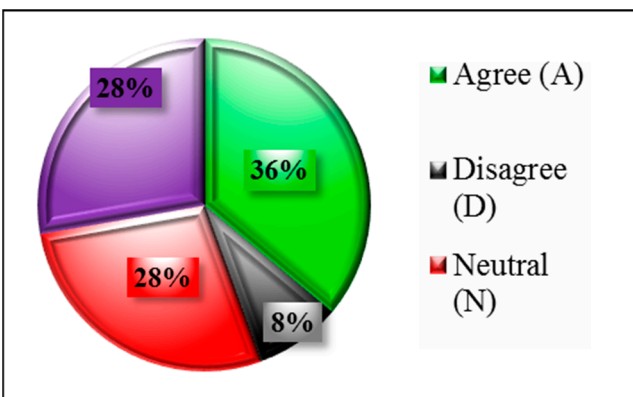

**Figure 4.** Client presence in the process dimension of the agile process.

A full set of accurate, agile practices in the technical component of agile processes leads to a wonderful working project, according to Figure 5. The remaining 2.78% disagreed, showing that the agile approach's technological aspects may assist a project to achieve its goals. Finally, the result addresses the project management gap that causes delays. Using a full set of agile approaches, the project may be segmented and the required quantities, resources, and prices determined. It may then be delivered implant-approved to reduce errors, project delays, and completion time.

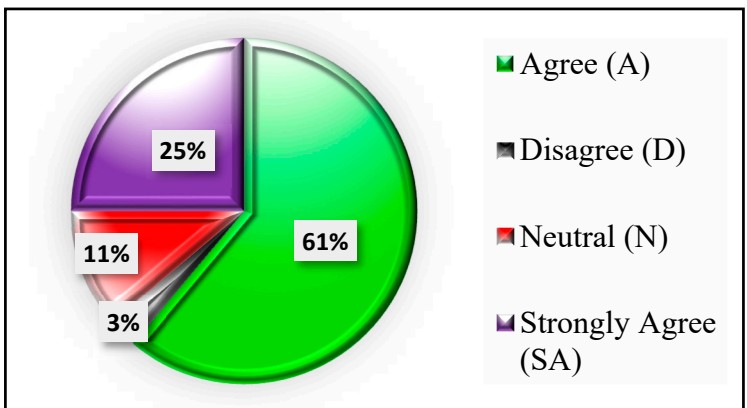

**Figure 5.** Complete set of precise, agile practices in the technical aspect.

With 13.89% unsure and 2.78% disagreeing, 83.33% agreed that risk management analysis is important and influences costs and efforts (Figure 6). According to the data, avoiding risk detection and mitigation is one of the key project management shortcomings. In order to reduce the project's hazardous effects, the project manager must prepare a risk analysis chart containing impact and probability analysis. Figure 7 reveals that 82.5% of respondents feel that management creates a clear plan for team members to understand the project's long-term vision and goals, while 7.5% are unsure. The remaining 2.5% disagree. According to the research, a lack of vision and clarity may limit the project's scope and make it harder to define deliverables.

Figure 8 indicates that 70% of the respondents have agreed that lean methodology helps in proper project scheduling, whereas 20% have remained neutral; the remaining 10% have disagreed with the statement.

Figure 9 demonstrates that 77.5% of respondents say management regularly investigates and analyzes the major project approach to improve assessment, control, and risk management; 7.5% of respondents disagree, whereas 15% are open-minded and indifferent. The majority agreed, indicating the benefits of project review and monitoring.

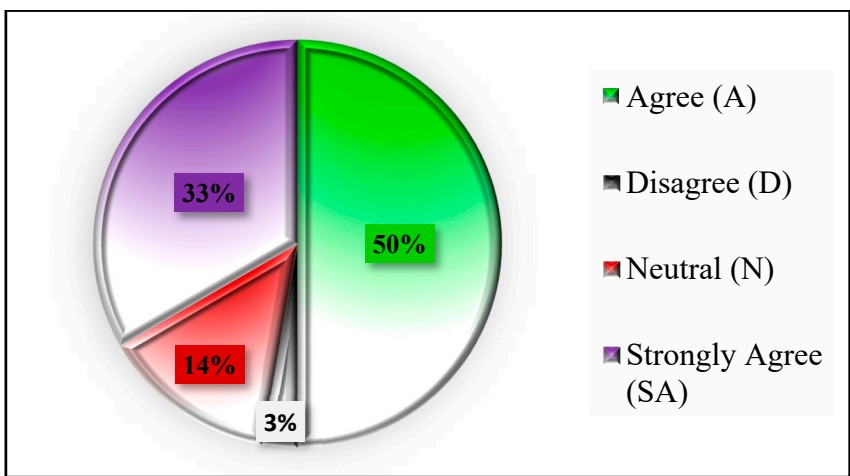

**Figure 6.** Risk analysis and mitigation in project management.

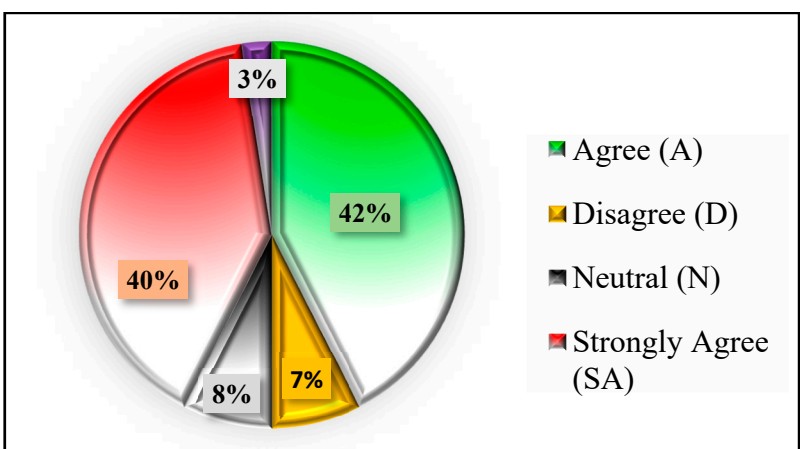

**Figure 7.** Vision and clarity in project management.

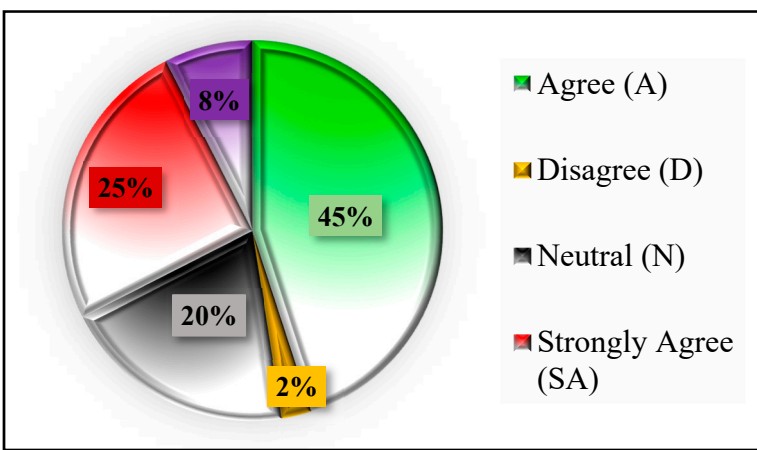

**Figure 8.** Proper scheduling under the lean methodology.

Figure 10 reveals that 88.88% of respondents stated that teamwork and customer ties influenced project delivery; 5.56% were neutral, while 5.56% disagreed, showing that effective delivery necessitates customer relationships. This figure illustrates how collaboration improves project worker productivity and motivation, as well as customer interactions. The lean strategy ensures this by including all project members in the design and execution, as well as the principal customer or end-user.

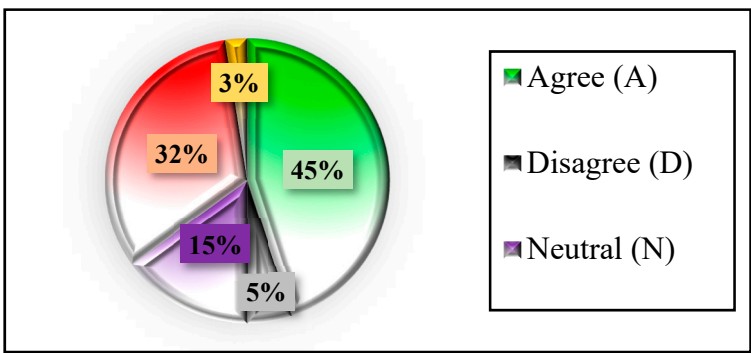

**Figure 9.** Review of project procedures for project management.

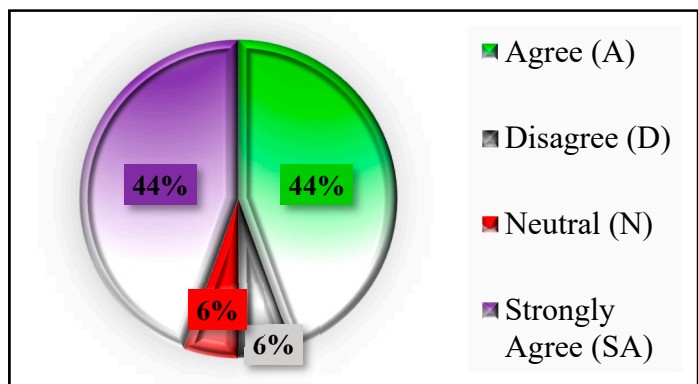

**Figure 10.** Teamwork and customer relationship under the lean methodology.

*4.4. Correlation*

Correlation measures the strength between the variables and determines the significance level effectively. The correlation table below indicates that the value of variables is below 0.8. This shows a low level of correlation and variables. Therefore, there is no high significance between the variables. Referring to Table 8, it can be inferred that the relationship between the selected variables is positive.

**Table 8.** Agile correlation between the selected variables.

| Variables | Implementation | Commitment | Team Work | Progress Tracking | Client Presence | Agile | Completion |
|---|---|---|---|---|---|---|---|
| Commitment | 0.535 | | | | | | |
| Team Work | 0.282 | 0.282 | | | | | |
| Progress Tracking | 0.476 | 0.31 | 0.202 | | | | |
| Client Presence | 0.161 | 0.229 | 0.15 | 0.288 | | | |
| Agile | 0.729 | 0.664 | 0.239 | 0.38 | 0.288 | | |
| Completion | 0.417 | 0.479 | 0.297 | 0.016 | 0.411 | 0.459 | |
| Risk Analysis | 0.269 | 0.295 | 0.221 | 0.514 | 0.44 | 0.215 | 0.201 |

The correlation shown in Table 9, indicates that the value of variables is below 0.8. This shows a low level of correlation and variables. There is no high significance between the variables. This indicates that the value of 0.8 is the threshold, and a value falling below 0.8 suggests that the results of the lean methodology are not significant. The results and findings state that these approaches made managing project tasks efficient and practical and ensured swift decision-making.

**Table 9.** Lean correlation with variables.

| Variables | Lean Method | Observe | Equipment | Definition of Management SOPs | Formulation of Management for Plan | Lean Method |
|---|---|---|---|---|---|---|
| Observe | 0.324 | | | | | |
| Equipment | 0.475 | 0.144 | | | | |
| Management to ensure task | 0.164 | 0.289 | 0.253 | | | |
| Management formulates a specific plan | 0.003 | 0.529 | 0.135 | 0.236 | | |
| Lean method | 0.187 | 0.246 | 0.154 | 0.275 | 0.241 | |
| Management | 0.176 | 0.312 | 0.206 | 0.33 | 0.314 | 0.271 |

### 4.5. Two-Sample t-Test Analysis

The gender *t*-test shown in Figure 11, the null hypothesis of the analysis is 0, while the alternative hypothesis is not equal to 0. The *t*-value of the Null hypothesis is 0.02, having a degree of freedom of 34. In comparison, the *t*-value of the alternative hypothesis is −0.64, with a DF value of 38.

| | | |
|---|---|---|
| Null hypothesis | | $H_0: \mu_0 - \mu_1 = 0$ |
| Alternative hypothesis | | $H_1: \mu_0 - \mu_1 \neq 0$ |
| **t-Value** | **DF** | **p-Value** |
| 0.02 | 34 | 0.986 |
| Null hypothesis | | $H_0: \mu_0 - \mu_1 = 0$ |
| Alternative hypothesis | | $H_1: \mu_0 - \mu_1 \neq 0$ |
| **t-Value** | **DF** | **p-Value** |
| −0.64 | 38 | 0.528 |

**Figure 11.** Gender *t*-tests.

Referring to Figure 12; both cases of Agile and Lean, respectively, the *p*-value of the *t*-test is higher (*p*-0.986, *p*-0.528 > 0.05) than 0.05, which states that samples are typically distributed, and there is a difference or variance between the population mean. This signifies that the opinions of males and females regarding the effectiveness of Agile and Lean approaches are the same. Since it is statistically insignificant, the value indicates that our postulated null hypothesis is true since the *t*-tests are not low enough to reject it, i.e., 0.528 and 0.986.

| | | |
|---|---|---|
| Null hypothesis | | $H_0: \mu_0 - \mu_1 = 0$ |
| Alternative hypothesis | | $H_1: \mu_0 - \mu_1 \neq 0$ |
| **t-Value** | **DF** | **p-Value** |
| 0.02 | 34 | 0.986 |

**Figure 12.** Age *t*-tests.

In the case of Agile, it can be stated that the *p*-value is higher, that is 0.986, and this indicates that the samples are typically distributed; there is a slight difference between the variables. The opinions of all age groups are similar regarding the agile approach.

Referring to Figure 13, it can be stated that the *p*-value is higher than 0.528, which indicates that the samples are normally distributed, and there is little difference between the variables.

| Null hypothesis | | $H_0$: $\mu_0 - \mu_1 = 0$ |
|---|---|---|
| Alternative hypothesis | | $H_1$: $\mu_0 - \mu_1 \neq 0$ |
| *t*-Value | DF | *p*-Value |
| −0.64 | 38 | 0.528 |

**Figure 13.** Lean *t*-tests.

The opinions of all age groups are similar regarding the lean approach. Since the significant value is more than the standard 0.05, i.e., 0.528, the data has a strong likelihood of having a null hypothesis and thus cannot be rejected for the entire population.

*4.6. One-Way ANOVA*

Table 10 prsents the analysis of variance which indicates that the *p*-value of the conflict is 0.588, which is higher than 0.05. This suggests no significance between age and agile methodology as most of the respondents have similar views regarding agile methods.

**Table 10.** Analysis of variance of age.

| *Source* | *DF* | *Adj SS* | *Adj MS* | *F-Value* | *p-Value* |
|---|---|---|---|---|---|
| **Age** | 3 | 1.139 | 0.3798 | 0.65 | 0.588 |
| **Error** | 32 | 18.654 | 0.5829 | | |
| **Total** | 35 | 19.793 | | | |

Table 11 shows mode summary of the ANOVA indicates that the R-square value is 5.76%, which shows that the model is predictable by 5.76%. Therefore, from the R-square value, it can be interpreted that the selected ANOVA model for analysis suitably fits with the collected data and thus is regarded to be a better fit for the model. Moreover, it increases the overall prediction of the model as well.

**Table 11.** Model summary of the ANOVA.

| *Scheme* | *R-sq* | *R-sq (adj)* |
|---|---|---|
| 0.763493 | 5.76% | 0.00% |

Figures 14 and 15, shows the gender responses as it relates to lean and agile methodologies respectively. It is clear that male and females have different perspectives on lean and agile while each group has the same level of understanding of each topic.

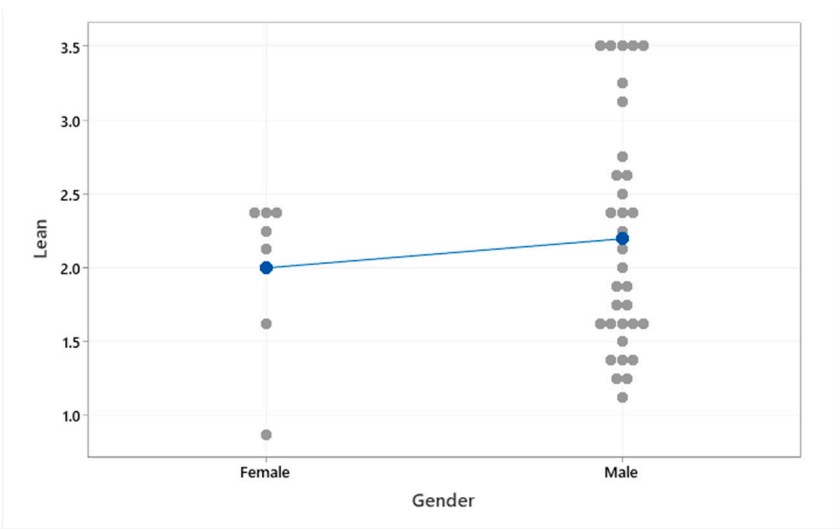

**Figure 14.** Individual value plot of lean vs. gender.

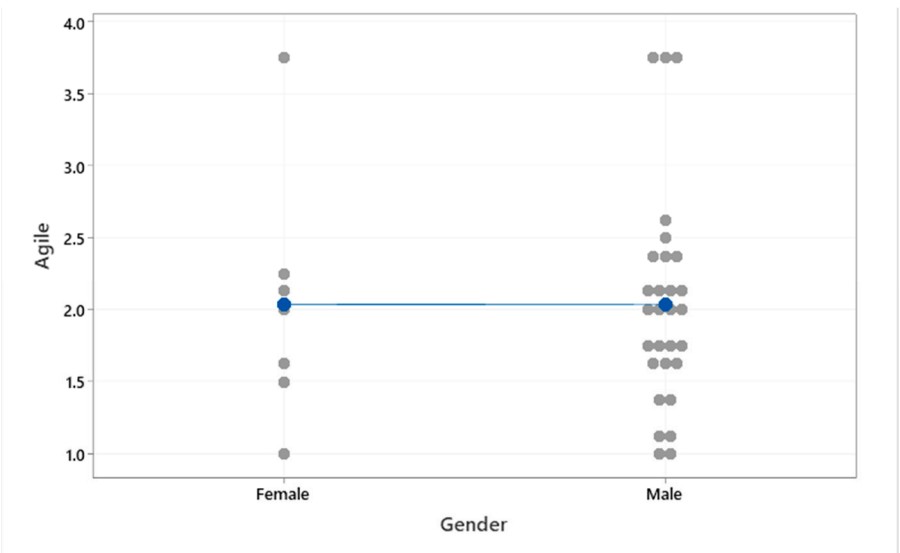

**Figure 15.** Individual value plot of agile vs. gender.

Figure 16 indicates that the responses of participants of most respondents are almost the same regarding agile methodology; however, the answers have changed in the age group of 61 to 70.

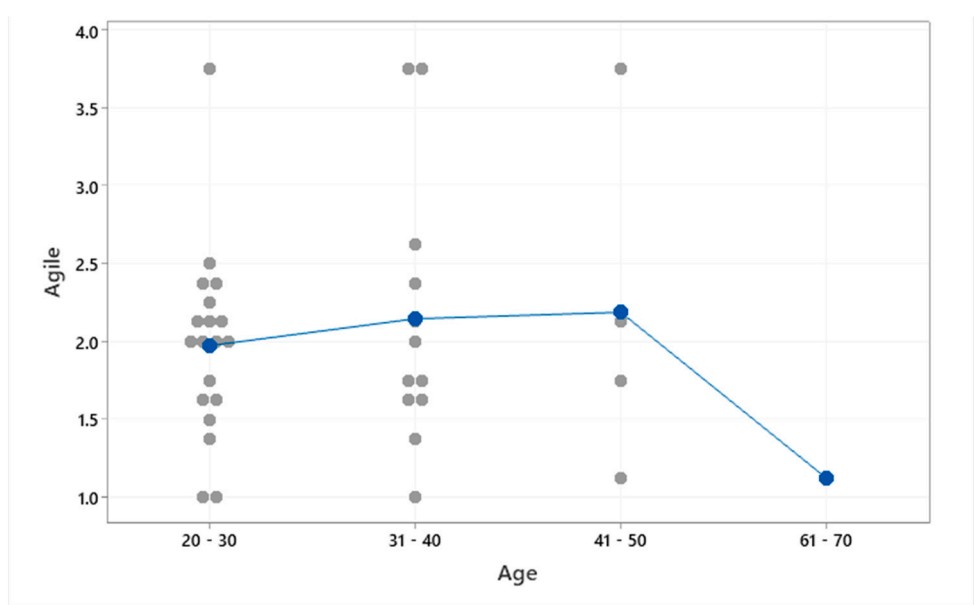

**Figure 16.** Individual value plot of agile vs. age.

Table 12 indicates that the F-value of the conflict is 0.97, which is higher than 0.05. This suggests no significance between age and lean methodology as most of the respondents have similar views regarding lean methods.

**Table 12.** Analysis of variance between lean and age.

| Source | DF | Adj SS | Adj MS | F-Value | p-Value |
|--------|----|--------|--------|---------|---------|
| **Age** | 4 | 2.117 | 0.5293 | 0.97 | 0.436 |
| **Error** | 35 | 19.076 | 0.545 | | |
| **Total** | 39 | 21.194 | | | |

Table 13 indicates mode summary of the ANOVA, R-square value is 9.99%, and this shows the model is predictable by 9.99%. Since the *p*-value is 0.436, the relationship is statistically insignificant, suggesting a marginal association between age and lean methodology.

**Table 13.** Model summary.

| S | R-sq | R-sq (adj) |
|---|---|---|
| 0.738268 | 9.99% | 0.00% |

Figure 17 indicates that the responses of participants are almost the same regarding lean methodology. However, the answers have changed in the age group of 61 to 70.

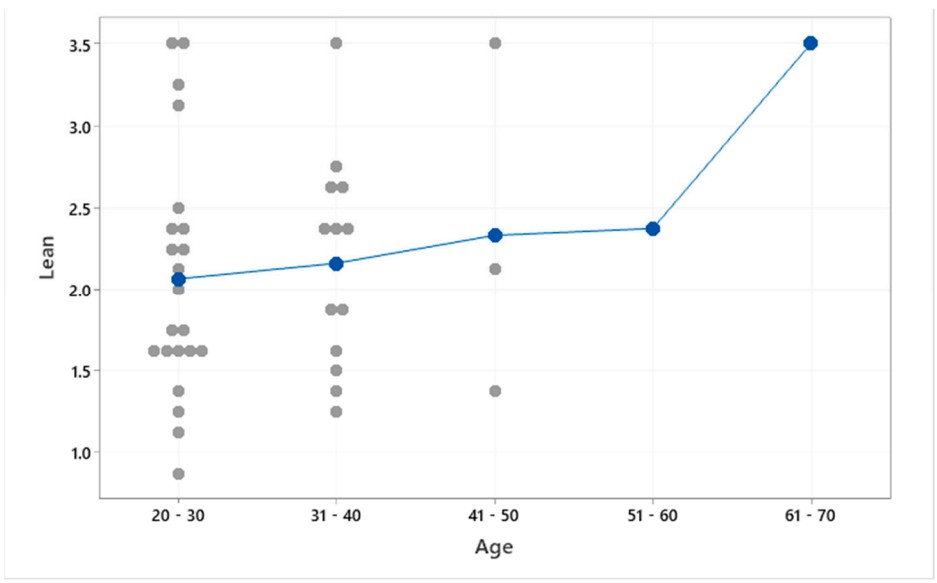

**Figure 17.** Individual value plot of lean vs. age.

## 5. Discussion

The findings of the study were discussed, and it was determined that Agile and Lean approaches are helpful for software development firms, startups, and large corporations. Twenty years after the Agile Manifesto was published, this method has been shown to be useful for many businesses and is now being implemented by a large number of for-profit organizations. The discussion showed, however, that many non-profits are still unfamiliar with the concept of agility. The implementation of agile approaches and choices that are based on related ideas, such as design thinking and deliver significant benefits are straightforward in this environment. In the case of transnational non-profit organizations, the challenge is made much more problematic by the existence of governmental regulatory bodies, stakeholder partners, references, rules, and other aspects that are difficult to define.

In contrast to the business sector, communication between different levels of non-profit organizations, funders, local partners, and project beneficiaries is very difficult and cannot be streamlined or expedited as in the corporate sector. Development is one of the new ways in which individuals and organizations in the area attempt to provide a helping hand. This study on the relevance of lean and agile projects in non-profit organizations has helped to gain a better understanding of the forces that work against project acceptance. Furthermore, nonprofit managers may obtain an understanding of how these programs affect the functioning of their organizations as well as the factors that determine the effectiveness of their programs. There are various misconceptions and reasons why nonprofit organizations should not consider adopting old and new techniques for operations management, as well as reasons why they should explore doing so.

Companies that are not in the manufacturing industry, such as non-profit organizations, are erroneous examples of corporations that should not be embracing lean and agile approaches. However, in order to execute Agile and Lean methodologies, large resources are required. Nevertheless, this is a transitory trend since the enterprises involved are small. They are unable to plan effectively using Agile and Lean approaches due to a shortage of time. The present study raises awareness of Agile and Lean approaches as production techniques, and it reduces the inconsistency of these approaches when used as such. Specifically, the relevance and benefits of such initiatives are highlighted in this study.

Companies that have customers, do the same things over and over and want to improve their performance may be able to work more quickly and effectively to be more efficient and make more money. Participants' private and sensitive information is not shared with a third party or used for any purpose other than this study, as required by ethical guidelines. In addition, all participants are supplied with an informed consent form, which they must sign in order to confirm their participation in the study. According to the findings, no volunteer had been harmed or pressured into taking part in the research.

Further, the results also indicated that agile methodology in project management and team commitment is significantly correlated (Pearson 0.664); this implies that one of the critical project management gaps in NPOs is team commitment because the nonprofit project is addressed by an agile methodology that is encouraging team commitment and motivation. These results contradict the discussion where the researcher argued that, conversely, team commitment is lacking in agile methods and, therefore, cannot be applicable for projects where team commitment is already an issue. The fundamental tenet leads teams to commit to projects incrementally, as they better comprehend the work by applying it in small chunks.

This study also signifies that agile methodology has a low correlation with client involvement, 0.288. This could be a significant major project management gap that agile methods cannot address. This is because the client's participation is massively crucial for the project to be successful. After all, the client is the core focus point that describes the project objectives, requirements, and deliverables; if the client is not involved while planning and initiating the project, this could be a significant failure.

On the other hand, it has been analyzed that lean methodology is ineffective in addressing project management gaps concerning project progress observation, equipment management, and risk management. The results indicate that the lean method has a weak correlation with equipment management, whereas the Pearson correlation is ($p$-0.154). The results are in contradiction with the findings where the researcher argued that control of equipment is one of the main requirements for practical project handling; where effective equipment and resource management leads to less waste generation, makes the project sustainable, and saves costs, which prevents cash deficit situations in the long run.

## 6. Conclusions and Recommendations

This study explores how Agile and Lean approaches will be employed in non-profit organizations' financial and non-financial operations. This study will also help non-profit organizations by showing them how to use Agile and Lean approaches to better their operations. Only a few studies have looked at the financial and non-financial effects of speed and autonomy on nonprofit organizations. This study fills certain gaps in the previous literature. In the past, non-profits have looked at just speedier and less successful strategies, such as Agile and Lean. This research shows how to plan to use Agile and Lean in non-profit organizations to reduce risk and costs. Prior research has shown why Agile and Lean did not work for charities.

A nonprofit organization's success is frequently measured by how well it achieves its goal, rather than how much money it makes. This study examines how Agile and Lean approaches may enhance a nonprofit's financial and non-financial performance. This study will help the industry by identifying how non-profits seek to enhance their financial and non-financial performance. The study looks at how Agile and Lean structures and

technology help projects in the nonprofit sector be successful. From a senior management perspective, the research highlights the benefits of building strong non-profits. The study also revealed new connections between different performance management systems that encourage less robust and faster techniques.

The research indicated that speed may be the best choice for changing systems or solving difficult problems. Agile and Lean will work well with project management and administration, as well as an understanding of how to alter and implement swiftly. This investigation's only flaw is time limits. Future academics may use longitudinal methodologies to investigate study concerns and identify sustainable non-profit income growth strategies. The study also suggests additional research on non-profit organizations' failures and continuous improvement initiatives utilizing Agile and Lean methodologies. Participants in the study thought that communication was very important. Because of this, future research should focus on improving NPO communication while also trying to obtain more people to work for them and help as volunteers.

Finally, the Agile and Lean methodologies for closing project management gaps in nonprofit organizations have been thoroughly investigated and evaluated. Furthermore, the conversation has been considered compatible with the study's objectives. The findings show that rising project complexity requires the creation of specialized project management techniques capable of effectively and efficiently executing complex projects in non-profit organizations.

**Author Contributions:** Conceptualization, A.A.; Formal analysis, F.A.; Investigation, F.A.; Methodology, F.A.; Software, F.A.; Validation, F.A.; Writing—original draft, F.A.; Writing—review and editing F.A. and A.A. All authors have read and agreed to the published version of the manuscript.

**Funding:** This research received no external funding.

**Institutional Review Board Statement:** The study did not require ethical approval.

**Informed Consent Statement:** Not applicable.

**Data Availability Statement:** Not applicable.

**Acknowledgments:** The authors gratefully acknowledge the support provided by the Master of Engineering Management Program at Prince Sultan University (PSU).

**Conflicts of Interest:** The authors declare no conflict of interest.

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
