# Peer review of "A Critical Analysis of Agile and Lean Methodology to Fulfill the Project Management Gaps in Nonprofit Organizations (NPOs)"

_applsci, doi:10.3390/app12115467_

Round 1
Reviewer 1 Report
Review Report
A Critical Analysis of Agile and Lean Methodology to Fulfill the Project Management Gaps in Nonprofit Organisations (NPOs)
This research examines the implications of lean and agile methodologies for solving project management gaps in Nonprofit organizations (NPOs). This research provides a new avenue for scholars in the context of Agile and Lean Methodology to Fulfill the Project Management Gaps in Nonprofit Organisations (NPOs) knowledge. However, this research has the following drawbacks that need to be addressed carefully with logical and theoretical justification despite its novelty. The following corrections must be made to improve the quality of this research.
Introduction
Overall Well written, but Please discuss your problem statement and justify how it can be solved. Please state the key objectives of the research study and how they can be answered.
Literature Review
This part must discuss each variable and previous work done. This paper very briefly highlighted previous studies on Agile and Lean Methods to Fulfil the Project Management Gaps in Nonprofit Organisations (NPOs). Please expand and discuss in detail.
We mostly justify the variables or model literature and theories. So authors should briefly explain the literature gap and add a section on relevant theories.
Methods, data collection, population, and sampling
The authors mentioned, Data was distributed among managers, and 97 participated? It is not clear how much sample was chosen from the total population on which bases? Further, which type of sampling technique was applied, probability sampling. Authors must discuss and justify how sampling techniques are employed in the methods part. T-test and ANOVA were used. Justify why these technique were used. We normally decide based on the type of research and nature of the data. i.e., either continues or discrete. The data analysis showed the data scale was 5 point Likert ( 1= strongly disagree to 5= strongly agree ); why not this study opt for other analysis tools.
The authors mentioned that “The data was gathered from secondary sources while the researchers gathered the 161 original data” this research data is secondary or primary ? Normally we use survey questionnaires for collecting primary data? The questionnaires used for data collection were adopted from previous studies or developed by the authors. Please clarify and justify.
Data Analysis
The data analysis looks fine.
Discussion and findings
Part 5 should be titled “Discussion and Findings”
Theoretical and empirical discussion is ok but must highlight the managerial and practical aspects in detail.
Conclusion and recommendations
The limitations part is not highlighted before proceeding with future call, and please incorporate it either in this part or end of the discussion part.

Reviewer 2 Report
Review for
A Critical Analysis of Agile and Lean Methodology to Fulfill the Project Management Gaps in Nonprofit Organizations (NPOs)
The level of originality of the paper is high. The literature review and proposed methodology are properly discussed and compared to the previous studies.
In this paper, authors used any sources, containing both historical and fundamental works, as well as the latest scientific research on this topic. But the literature review can be structured. The papers discussed many points of this study. Please, discuss these papers:
Moiseev, N., Mikhaylov, A., Varyash, I., Saqib, A. (2020). Investigating the relation of GDP per capita and corruption index. Entrepreneurship and Sustainability Issues, 8(1), 780-794. http://doi.org/10.9770/jesi.2020.8.1(52)
Dinçer, H., Yüksel, S, Mikhaylov A., Barykin S.E., Aksoy T., HacioÄŸlu U. (2022). Analysis of environmental priorities for green project investments using an integrated q-rung orthopair fuzzy modelling. IEEE Access, 10, https://doi.org/10.1109/ACCESS.2022.3174058
Kranina E.I. (2021). China on the way to achieving carbon neutrality. Financial Journal, 13, 5, 51–61.
Bushukina V. (2021). Specific Features of Renewable Energy Development in the World and Russia. Financial Journal, 13, 5, 93-107.
Matveeva N. (2021). Legislative Regulation Financial Statement Preparation by Micro Entities: International Experience. Financial Journal, 13, 5, 125-138.
The introduction section has benefit from having a clearer structure of what to expect in the paper. Furthermore, the author(s) would benefit from being more concise in their writing, as much of the content was redundant and overemphasized. While it is good practice to assume the reader has no prior knowledge of the content, a topic and/or discussion does not need to be explained over and over again if it is stated both adequately and appropriately once.
Some conclusions contribute to the study of the problem. The author does not formulate the problem itself – it makes impossible to analyse the contribution of the paper. The aim or the question of the paper (or even the hypothesis of the author) are formulated.
Overall, it is very clear to grasp understanding of the manuscript and content in its current state. I strongly advise using hypothesis points to articulate and/or express material in scientific writing. Publication of this piece seems likely in any reputable scientific periodical after a correction in the writing of the manuscript.
Table 1. 2020 GIBS Survey Country and Group Details is important to explore the specifics. Some new groups can be contributed to this table to the study of the problem.
Authors need to add more details on the range of simulation considered in this work should be clearly outlined within the abstract. The current statements are vague and too general to get an idea of the work that have been accomplished.
The paper possesses a proper form of well-structured and readable technical language of the field and represents the expected knowledge of the journal`s readership.
There are minor errors in English, but this does not affect the general nature of the work. The current study brings many new to the existing literature or field. For one, the author(s) seem to have a good grasp of the current literature on their topic area (i.e., recent literature and seminal texts relevant to their study is not cited/referenced).
Reviewer 3 Report
The research question is topical and brings a novel approach regarding the real impacts agile and lean strategies for project management gaps in the NPO.
The Title – is rather long but well focused on the content of the paper.
Abstract: Even if it is brief, the abstract has a standard formulation and presents concisely the purpose, the method, the content, and the results of the research.
LINE 18 ;; ERROR : requiinvolvesojeing , andct
KEYWORDS – NOT accordingly. There are too many key words and key phrases.
1. The Introduction: is too long and implies also Literature Review, which should be a distinct section.
2. Review of literature
Literature review should be enriched in references for supporting the same concept or idea/ or theory. This section is very short – and this results also in a short list of References .
Both Literature review and References list should be improved and increase references on the topic.
3. Material and Methods
The name of the section – does not correspond to the content … Would rather change into RESEARCH DESIGN..
It is too brief, and should be better incorporated together with Research design.
4. Results
Research methodology is not very well presented, and results are separated into many sections which are not clearly connected.
Hypothesis and objectives are not stated from the start of the paper, thus these are not well argued throughout literature and after that validated through measurements.
The presentations of results is low level.
5. Discussion
In the Discussion section are presented some ideas that are simple assumptions without any clear evidence or support from the research data.
Discussions should be improved and provided with numbers and data from the research.
6. Concluions.
Conclusions and recommendations are formulated very general and don’t reflect well enough the contributions of this research.
The authors should connect better the final ideas to the general goal of the paper and to the practical achievements of the research.
References are incomplete and should be generously improved.
Author Response
Please, see the attachment.
Regards

Reviewer 4 Report
Topic
The topic is long. It does not excite the reader.
Introduction
There is an identified question.
More emphasis can be placed on the research question of the study.
This should happen after the gap in the literature has been identified.
Literature Review
The literature review section is insufficient. The literature review of the topic should be better supported by reference and current studies.
It does not make sense for the introduction section to be larger than the literature review section.
Methodology
The methodology section needs to be improved.
There needs to be more depth in the information provided to the reader that allows for replication of the study.
Results and Discussion
There is a results section. The discussion could strengthen the results obtained in comparison with the literature.
Conclusions and Recommendations
The conclusions are in line with the study.
Recommendations are made for future research.
Round 2
Reviewer 3 Report
I appreciate the authors' effort to improve the paper. It is now ready for publication.